# The Strategies of Development of New Non-Toxic Inhibitors of Amyloid Formation

**DOI:** 10.3390/ijms24043781

**Published:** 2023-02-14

**Authors:** Oxana V. Galzitskaya, Sergei Y. Grishin, Anna V. Glyakina, Nikita V. Dovidchenko, Anastasiia V. Konstantinova, Sergey V. Kravchenko, Alexey K. Surin

**Affiliations:** 1Institute of Protein Research, Russian Academy of Sciences, 142290 Pushchino, Russia; 2Institute of Theoretical and Experimental Biophysics, Russian Academy of Sciences, 142290 Pushchino, Russia; 3Institute of Environmental and Agricultural Biology (X-BIO), Tyumen State University, 625003 Tyumen, Russia; 4Institute of Mathematical Problems of Biology RAS, The Branch of Keldysh Institute of Applied Mathematics, Russian Academy of Sciences, 142290 Pushchino, Russia; 5Faculty of Biotechnology, Lomonosov Moscow State University, 119991 Moscow, Russia; 6The Branch of the Institute of Bioorganic Chemistry, Russian Academy of Sciences, 142290 Pushchino, Russia; 7State Research Center for Applied Microbiology and Biotechnology, 142279 Obolensk, Russia

**Keywords:** amyloid, α-synuclein, insulin, amyloid-beta, Alzheimer’s disease, Parkinson’s disease, diabetes

## Abstract

In recent years, due to the aging of the population and the development of diagnostic medicine, the number of identified diseases associated with the accumulation of amyloid proteins has increased. Some of these proteins are known to cause a number of degenerative diseases in humans, such as amyloid-beta (Aβ) in Alzheimer’s disease (AD), α-synuclein in Parkinson’s disease (PD), and insulin and its analogues in insulin-derived amyloidosis. In this regard, it is important to develop strategies for the search and development of effective inhibitors of amyloid formation. Many studies have been carried out aimed at elucidating the mechanisms of amyloid aggregation of proteins and peptides. This review focuses on three amyloidogenic peptides and proteins—Aβ, α-synuclein, and insulin—for which we will consider amyloid fibril formation mechanisms and analyze existing and prospective strategies for the development of effective and non-toxic inhibitors of amyloid formation. The development of non-toxic inhibitors of amyloid will allow them to be used more effectively for the treatment of diseases associated with amyloid.

## 1. Introduction

The aggregation of proteins and peptides into amyloid structures is one of the most intensively studied biological phenomena at the moment. Neurodegenerative diseases associated with the misfolding of proteins and the accumulation of such proteins in tissues are among the most prevalent in terms of social costs in the world, especially in developed countries. As medical advances and life expectancy increase, these diseases are also becoming a serious social factor in developing countries. Particularly relevant in this regard is the study of protein misfolding, which leads to the formation of pathogenic amyloid fibrils, with the deposition of which in tissues is currently associated with more than 30 human diseases [1]. The most common neurodegenerative disease is AD. According to the International Epidemiological Study of AD, in 2010, there were 35.6 million patients with AD in the world; this number doubles every 20 years, and it will reach 65.7 million by 2030 [2]. In America, among people over the age of 85, the prevalence of this disease reaches approximately 50%.

The second most common neurodegenerative disease is PD. In this case, pathological aggregation of α-synuclein occurs in the dopaminergic neurons of the substantia nigra with the formation of Lewy bodies [3]. At the same time, an increase in the level of α-synuclein in the cerebrospinal fluid is a marker of AD [4]. Additionally, α-synuclein is found in presynaptic cholinergic nerve endings that degenerate in the early stages of the disease [5].

Epidemiological, clinical and experimental evidence suggests an association between type 2 diabetes mellitus and AD [6]. Insulin modulates the metabolism of beta-amyloid precursor protein (APP) in neurons, reducing the intracellular accumulation of Aβ peptides, which play a key role in the pathogenesis of AD. Understanding the molecular mechanisms of this process (the mutual influence of insulin and Aβ peptide) underlies pathogenesis of the above two diseases. Aβ(1–42) and Aβ(1–40) have been shown to reduce the ability of insulin to bind to its receptors and also block receptor autophosphorylation. These data suggest that Aβ(1–42) and Aβ(1–40) are inhibitors of insulin receptor binding and function [7]. In essence, AD can be considered as one of the forms of “diabetes of the brain”, which contains both elements of insulin resistance and insulin deficiency.

The problem of finding effective inhibitors of protein and peptide aggregation is relevant both for the storage of appropriate drugs and for the treatment of diseases associated with amyloids. In modern preparations of insulin and its analogues, phenol and its derivative, metacresol, are usually used as inhibitors of the formation of amyloid fibrils and antimicrobial agents. Phenolic components in commercial insulin preparations also reduce the effectiveness of diabetes therapy due to the cytotoxic effect. Therefore, to solve the problem of the above-mentioned undesirable effects of the use of insulin preparations, it is necessary to find and propose new non-toxic inhibitors of fibril formation and preservatives that will replace toxic phenol and metacresol in insulin preparations and its analogues.

Despite intensive studies of the fibril formation of insulin, α-synuclein, and Aβ peptide (and other proteins and peptides), there is no generally accepted scheme for the formation of mature fibrils. The main difficulties arise in the interpretation of data on the onset of the fibrillation process. Different fibril morphology may result from similar but not identical pathways for the formation of mature fibrils. Differences can be laid at the beginning of the path of fibril formation, including at the nucleation stage. The question of the size of fibril nuclei formed by different proteins and peptides still remains open. Despite the large number of works in this area, there are still no reliable data on the mutual influence of insulin and the Aβ peptide, as well as the effect of modified forms of the Aβ peptide on the process of amyloid formation. In addition, there are no data on the causes and mechanism of the spontaneous deamidation of asparagine, which results in the formation of multiple forms of modifications of the Aβ peptide. It is known that these forms are able to accumulate in the human body as it ages, and, in all likelihood, are able to take an active part in amyloid formation. The study of the effect of the seed on the formation of fibrils is directly related to the molecular mechanisms of the development of amyloidosis. It is not yet clear how fibrils initially appear in healthy individuals. Seeds may play a key role in the development of systemic amyloidosis, and may also be an infectious agent involved in the development of prion diseases (or triggering prion disease). Knowing the size of the nucleus is necessary to stop further fibril growth and to search for pharmacoperones to correct misfolded protein forms.

An important biophysical difference between protein folding into a native structure and the formation of amyloid fibrils is that, during folding into a native structure, a correspondence between the amino acid sequence and the uniqueness of the folded state is assumed, whereas during amyloid formation, the same polypeptide sequence can form fibrils of different morphology. It is clear that knowledge of the causes and mechanisms of fibril formation is an important task not only for the treatment, but also for the prevention of various human pathologies, and also deepens our understanding of the nature of the formation of complex natural protein complexes.

Considering the fact that the reasons and, consequently, the methods of treating AD and PD have not yet been sufficiently studied, there are many approaches to studying these diseases, both in the context of computational research at the molecular level, and in the context of drug discovery for the treatment of this disease. Figure 1 demonstrates the PubMed statistics for three amyloidogenic proteins considered in this review.

This account reviews molecular mechanisms causing amyloidogenesis and recent prospective strategies for the search inhibitors of amyloid formation for three amyloidogenic peptides and proteins: Aβ peptide, α-synuclein, and insulin.

## 2. Amyloid Fibril Formation Mechanisms and Fibril Formation Nucleus Size Estimation

In the past decades, many studies have focused on amyloid proteins, which work as building blocks of toxic aggregates and expedite fibrillization. How various amyloid diseases are caused by the accumulation and deposition of protein oligomers and fibrils is yet debatable; these possible structures were suggested by MD simulations, in vitro, and in vivo experiments [8]. To date, several models of amyloid formation have been proposed, which allow, based on kinetic data on aggregation, to determine which mechanism is implemented in the current experiment [9,10,11,12,13]. Various kinetic models of the process of amyloid formation were proposed. Several conclusions were made: (a) fibril formation is a process with an energetic barrier and therefore is dependent on the nucleation stage (formation of the most unstable structure); (b) simple model of amyloid growth in which seeds—the least stable structures—are enlarging by incorporating monomers by the points of growth (since amyloid are of a rod-like shape, it is safe to assume that amount of points of growth is constant per seed); this is not adequate in terms of fitting of this model to the solid share of experimental data.

The fibrillation process can be divided into three main stages, shown in Figure 2. First, primary nucleation, in which aggregates are formed directly from the monomer. Second, the fibril elongation step, where new monomers from solution are added to an existing aggregate by interacting with its free ends. Therefore, the rate of fibrillation is directly proportional to the number of free ends. Thirdly, the stage of “reproduction”, i.e., an increase in the number of aggregates due to the formation of secondary nucleation nuclei on the surface of existing aggregates or the breakdown of fibrils into oligomers and monomers, which act as a “seed” for amyloidogenic aggregation of monomers, which leads to the gradual accumulation of toxic aggregates and aggravation of diseases [14,15]. Large, stable oligomers are well suited as seeds [16,17,18].

The current view on the process of amyloid formation states that growth regimes following the nucleation stage can be divided into two types: the “linear” growth regime of protofibrils, where the possible number of points of growth (the place where monomers can attach to the fibril) is proportional to the number of nuclei (and as mentioned before such model is not applicable in general), and the “exponential” growth regime, where the possible number of points of growth during aggregation can greatly exceed the number of nuclei, i.e., growth process is somehow accelerating.

An analysis of the kinetic curves of amyloid formation revealed that it is often the exponential growth regime that is happening in experiments. There are various mechanisms which might accelerate fibril growth, but in general, all the scenarios can be roughly reduced to two scenarios (and their derivatives)—fragmentation and growth from the surface. In the case of fragmentation, the number of growth points increases due to the breakdown of fibrils. In the case of a bifurcation, deformations or some imperfections of the fibril surface might become new growth points, or be a trait for the secondary nuclei formation, which then will act as new growth points.

It is safe to say that nowadays, theoretical findings allow one to determine in which case it is possible to apply the linear growth model to the current experiment data, and in which case exponential growth models should be considered (Figure 3). Analysis has shown the relation between several characteristic values (a lag time T_lag_, time of transition from monomers to the fully aggregate state T_2_ and their relation L_rel_ = T_lag_/T_2_) to be critical while choosing a model. Turns out that L_rel_ is critical in both distinguishing linear growth from exponential and in the process of estimation of the sizes of primary nucleation during exponential growth. Based on the obtained relationships between the characteristic times, it was concluded that it is necessary to perform a series of kinetic experiments, where the monomer concentration would be the only variable. Such experiments do allow one to determine the sizes of a nuclei both primary and secondary (if secondary nucleation takes place). Unfortunately, there is ambiguity in exponential growth scenarios and direct estimation of a mechanism of exponential growth cannot be ruled out on the basis of relation between characteristic times. Therefore, to determine the mechanism one should address the extra experiments, the interpretation of the results of which is able to unambiguously determine the mechanism. At the same time, the relation between quantities L_rel_, T_lag_, T_2_ might give an insight and therefore narrow down the range of possible mechanisms in some cases.

## 3. Protein Folding and Aggregation

### 3.1. Aβ Peptide

One of the factors leading to the death of nerve cells and cognitive impairment that accompanies AD is the pathological accumulation of Aβ peptide aggregates in the brain tissue, which is the main component of senile plaques and characteristic morphological features of AD. The Aβ peptide, which is a polypeptide with a molecular weight of about 4 kDa and consisting of 40–42 amino acid residues, is formed from the Amyloid Precursor Protein (APP) [19]. Aβ has highly pronounced amyloidogenic properties, and its oligomers are toxic to nerve cells, causing neuronal degeneration and death [20,21]. Another characteristic morphological feature of AD is the disruption of the cytoskeleton of nerve cells and the accumulation of neurofibrillary tangles in them, which consist mainly of insoluble filaments of the hyperphosphorylated Tau protein [22,23,24,25]. It is believed that over the course of the neurodegenerative process, amyloid fibrils are first formed, which disrupt the functions of nerve cells, and then neurofibrillary tangles form in them.

Currently, there are three groups of enzymes (α, β, and γ-secretases) capable of cleaving APP with the formation of fragments of different lengths, which, in turn, have certain functional properties. From the point of view of drug targets in the treatment of AD [26], these secretases are being studied. As a result of the sequential action of β- and γ-secretases, sAPPβ, Aβ, and a short cytoplasmic fragment of AICD are formed. The products of APP cleavage by α- and γ-secretases are sAPPα, AICD and a fragment of about 3 kDa (p3) (Figure 4). The properties of p3 peptides consisting of amino acid residues 17–40 or 17–42 of the Aβ sequence are not yet known. In general, this part of the process of APP proteolysis, triggered by α-secretase, is considered non-amyloidogenic and does not play a significant role in the pathology of AD. About 90% of APP is cleaved along this pathway [27].

It should be noted that the fragment 17–40 or 17–42 of the Aβ sequence contains both known amyloidogenic regions, this does not prevent the formation of amyloid fibrils. Additionally, as we now understand, this is a conditional division into amyloidogenic and non-amyloidogenic pathways. To put it simply, judging by the assessment that 90% goes along the path where fragments 17–40 or 17–42 are obtained, then these are more important and necessary peptides for biochemical processes. Additionally, even now, we still do not fully know the function of the Aβ peptide itself.

Despite the large number of studies, only we, for the first time, have made calculations of the number of monomers included in the nucleus of the Aβ fibril, based on the model of amyloid formation developed by us [9]. In the case of Aβ(1–42), the size of the primary nucleus is three monomers, whereas the exponential growth process occurs according to the branching scenario, and the size of the secondary nucleus is 2 monomers [9,10,12]. For Aβ(1–40), these sizes are one less, the nucleus of primary nucleation is 2 monomers, and the nucleus of secondary nucleation is 1. The calculation was made according to the data obtained and published in the Dobson’s laboratory [11,28]. There are also no works where the spine of amyloid fibrils formed by these peptides has been determined using proteolysis and mass spectrometry. We have determined the spine of amyloid fibrils for Aβ(1–40) and Aβ(1–42) [16,18]. As a result, these data are consistent with the amyloidogenic regions predicted for the Aβ peptide. Additionally, all these data, along with data from electron microscopy and X-ray, made it possible to propose a model in which the fibril is built from oligomers consisting of at least 12 monomers [12,16].

The main building block of these fibrils is a ring-like oligomer with a diameter of about 8–9 nm, an internal cavity of about 2–3 nm and a height of about 2–3 nm. Fibrils are formed due to the interaction of such oligomers with the formation of fibrils up to several microns in length. At the same time, there are differences in the packing of ring oligomers in fibrils, which consists of their different ordering. A more ordered association of circular oligomers is observed for the recombinant Aβ(1–40) peptide. In addition, the oligomers in fibrils may interact with each other with a large overlap, which affects the increase in the width of the fibril in the places of their twisting. Additionally, the most disordered interaction of ring oligomers is observed in the synthetic preparation of the Aβ(1–42) peptide [16]. The polymorphism of this preparation is manifested only in different fibril widths. From the observational data, we conclude that the formation of these fibrils and their short fragments occurs through oligomeric ring-like structures [29,30]. The most remarkable thing about the morphology of both peptides, both recombinant and synthetic, is that at high magnification it can be seen that they are all formed from ring-shaped oligomers of approximately the same diameter, which reflects the diameter of the fibrils (Figure 5).

Only the formation of fibrils from oligomeric complexes can explain the polymorphism of amyloid fibrils. It must be remembered that the construction of actin filaments in a cell is served by about 150 proteins. That is, there is a whole machinery for building regular filaments. Additionally, the same should be applicable to the construction of any fibrillar structure. The structures obtained using cryo-electron microscopy are the result of mathematical processing, in which the end product should be the presence of “infinite” β-sheets. Additionally, it turns out that the results of electron microscopy contradict the results of cryo-electron microscopy [32]. A mixture of fibrils of different morphology is always present in solution and several different types of fibrils can be seen under the same conditions. Thus, the processing of amyloid fibrils using Markham’s rotation techniques [33] gives clear images of a fibril built from oligomers (Figure 6).

However, the question remains open, who builds such smooth structures in the cell? Additionally, we do not see such even and regular structures in any work. Why, having EM structures, no one wants to calculate X-ray reflections and compare them with the set obtained in the work of 1993 [34] and finally explain what the equatorial and meridional reflections at 55 and 53 Å mean, and, accordingly, other values.

For the last few years, several new structures of amyloid fibrils consisting of Aβ peptides have been obtained [35,36,37,38] (Figure 7 (6shs, 7f29, 6w0o, 7q4b, 7q4m)). The concept of fibril formation has not been changed at all: Aβ fibril has cross-β structure and consists of two stacks of peptide. The structures differ only in the conformation of the peptide in the stack and, consequently, in the region of interaction between the two stacks of peptides. In [35], the structure of amyloid fibrils consisting of the Aβ(1–40) peptide was obtained from the brain tissues of a patient suffering from Alzheimer’s disease (Figure 7 (6shs)). The authors of this work claim that this new structure has a right-handed twist and differs from the previously described structures obtained in vitro. Each individual peptide in this structure consists of four β-strands and has a C-shaped fold (Figure 7 (6shs)). In [38], the structure of amyloid fibrils consisting of the Aβ(1–42) peptide is presented, in which the amino acid residue Y10 is O-glycosylated. The Aβ(1–42) peptides in this structure have an S-shaped fold (Figure 7 (7f29)). Additionally, it differs from the previously obtained structures (for example, Figure 7 (2nao, 5kk3 and 5oqv)) by a conformation of 6–15 amino acid residues (Figure 7 (7f29)). Another new structure of amyloid fibrils consisting of the Aβ(1–40) peptide was obtained in the work [36]. A distinctive feature of this structure is that the peptides included in it have an elongated conformation (Figure 7 (6w0o)). In [37]. two types of fibrils consisting of the Aβ(1–42) peptide were obtained. In both cases, the fibrils have a left-handed twist, and the peptides that compose them are S-shaped (Figure 7 (7q4b and 7q4m)). These two types of fibrils differ in the number of β-strands in the peptide (in the first case, the peptide consists of five β-strands, and in the second—of four) and in the arrangement of the peptides relative to each other (in the first case, amino acid residues 36–37 lie inside the fibril, and in the second—25–26) (Figure 7 (7q4b and 7q4m)). Additionally, it was shown that the first type of fibrils in most cases is characteristic of patients with sporadic, and the second with hereditary and other types of AD.

### 3.2. α-Synuclein

α-synuclein is a protein expressed mainly in the presynaptic endings of the human central nervous system. Mutations in this gene change the structure of the protein, lead to its fibrillation, which is the cause of neurodegenerative diseases. Examples of synucleinopathies are PD, dementia with Lewy bodies, AD with Lewy bodies, and multiple system atrophy (MSA). All these diseases are characterized by the presence of amyloid inclusions in neurons, the main component of which is α-synuclein.

In the solution, the monomer does not have a clear secondary structure. The predicted disordered regions and structure are shown in Figure 8. The α-synuclein monomers are in the unfolded state in solution, and fast structural rearrangements of the molecules occur [49]. They constantly change their structure from globules to one- or two-tailed structures, which confirms that the protein is not structured [50]. α-synuclein does not form a stable tertiary structure and belongs to intrinsically unstructured proteins. However, aggregation results in the formation of amyloid fibrils consisting of β-sheets [51].

α-synuclein has several binding sites for metal ions. Thus, the second amino acid (aspartic acid) and region 48–53 (in particular, the fiftieth amino acid, histidine) are binding sites for the Cu^2+^ ion. Copper and iron ions promote α-synuclein aggregation and release of mature fibrils into the extracellular space, causing their further spread and enhancing the toxic effect [54]. The reason for the acceleration of aggregation upon binding of divalent metal ions is the change in contacts of the N- and C-termini to the NAC domain [55]. Triethylenetetramine (TETA) and deuterium fluoride (DF) can be used to inhibit the amyloidogenic effects of copper and iron ions. The action of these reagents illustrates the increased lifespan of *C. elegans*, a model of PD [54].

Amyloidogenic transformation of α-synuclein is the process of converting a protein from a disordered soluble monomer to insoluble fibrils containing many protein molecules thought to be organized in a cross-β structure. Such a structure implies the presence of β-sheets directed parallel to the fibril axis. In the course of amyloidogenic aggregation, various structures are observed: monomers that do not have a β-structure, oligomers, protofibrils, fibrils. In solution, all forms of α-synuclein exist in equilibrium, and the process of aggregation–disaggregation is constantly occurring [15]. Among all structures, oligomers and protofibrils are toxic due to the presence of a hydrophobic core and hydrophobic areas on the surface, which allow incorporation into the membrane [56] and the creation of a “pore” in it, whereas the membrane permeability for Ca^2+^ ions increases [57]. In addition, hydrophobic regions located on the surface promote self-aggregation of molecules due to hydrophobic interactions [58].

Other mechanisms of toxicity may include increased generation of reactive oxygen species, endoplasmic reticulum (ER) stress, inhibition of degradation in the proteasome, and mitochondrial dysfunction [59,60,61].

There is a variety of oligomers that differ in terms of molecular weight, the content of the β-layers, hydrophobicity, shape, and toxicity [62]. However, in terms of fibrillation, two types of oligomers are distinguished: “on-pathway”, which are directly converted into fibrils, and “off-pathway”, which are by-products of the reaction and remain in this form. Both the first and second types exhibit in vitro toxicity [63]. It has been shown that the addition of engineered oligomers to a primary culture of rat neurons leads to cell death [64]).

Mature α-synuclein fibrils are elongated aggregates consisting of two to four protofibrils [65]. The minimum number of monomers that make up a mature fibril is 70 monomers. Further addition of monomers leads to efficient fibrillation and replication [66]. The fibrils are folded into a “Greek key” structure. The core of this structure contains a lot of glycine, alanine, proline, i.e., small amino acid residues, which ensures a compact structure. The inner layer of the core is represented by amino acid residues 71–82 required for aggregation [67].

In the process of fibrillation in the human body, the deposition of Lewy bodies occurs. In the physiological state (monomer) of α-synuclein, neurons have a normal structure. At the oligomer stage, granularity appears in neurons, and with further protein folding into a fibril, Lewy bodies are formed with a compaction in the center, from which fibrils with a diameter of 10 nm diverge in different directions [62].

The rate of fibrillation depends on many factors such as pH, temperature, and protein concentration. For example, at low pH and an increase in temperature, aggregation proceeds much faster, which is associated with an increase in the overall hydrophobicity of the molecule and a decrease in the total charge in an acidic medium due to the protonation of carboxyl groups [58]. During the fibrillization of α-synuclein, protons released by the monomers are absorbed by the carboxyl groups of the C-terminus of the fibrils. In this case, the pH of the solution increases by an average of one [68].

Additionally, an increase in protein concentration enhances the process of fibril formation. The critical concentration for α-synuclein fibril growth is 0.4 μM. At this value, monomers can bind to the ends of fibrils, which leads to their growth. However, if only protein monomers are taken, then at least 10 µM of protein is required to start aggregation [69]. The calculated concentration of α-synuclein in the nerve cell is 70–140 μM. Thus, protein aggregation can start not only when it is overexpressed, but also under normal physiological conditions [65].

The pathological α-synuclein observed in synucleinopathies contains various post-translational modifications that have a direct effect on the fibrillation process. These include phosphorylation (90% of proteins in Ser-129), ubiquitination, acetylation, nitration, oxidation, end truncation [70]. Let us consider the effect of various modifications of α-synuclein on fibrillation. One known modification is the deletion of amino acid regions. Thus, comparing the processes of aggregation of a full-length protein and a protein with a truncated C-terminus, it turns out that the latter forms fibrils more easily [71,72]. Further study of this phenomenon showed that the full-length C-terminus prevents the aggregation of molecules due to the electrostatic repulsion of their negative charges [73].

N-terminal acetylation in nature plays an important function—it enhances protein–protein and protein-lipid interactions by neutralizing the charge of the N-terminus, which increases hydrophobicity. These data indicate that protein oligomers are stabilized [74]. A recent article shows different results. N-terminal acetylation reduces the rate of lipid-induced aggregation. Fibrils formed from acetylated α-synuclein have a different morphology compared to the unmodified protein. Thus, a reduced amount of β-structure was found in their structure [75].

Oligomers of α-synuclein are heteromeric structures differing in shape, structure, and functions. Using cryo-electron microscopy, two types of oligomers have been shown that have a similar cylindrical structure with a hole in the center [76]. The smallest, most stable 10S with an average molecular weight of 260 kDa and the larger 15S with an average molecular weight of 420 kDa. There is a direct correlation between the molecular weight of the molecules and the number of β-sheets: with an increase in the first, the second value increases. Thus, in 10S oligomers, 30% is in β-sheets, and in 15S, 40%. A similar correlation is found when measuring hydrophobicity: the maximum hydrophobic surface area is shown for large oligomers compared to other types of α-synuclein [77]. Oligomers are formed from antiparallel β-sheets [78], which allows them to integrate into the lipid membrane, forming pores in them and destroying cells due to ferroptosis [79]. Moreover, the addition of α-synuclein oligomers to the cell leads to the fact that, after their binding to the membrane, nucleation sites can form and the spread of α-synuclein will increase, stimulating apoptosis [80]. The inhibitor of this interaction of oligomers with membranes is squalamine, which changes the charge of lipids and blocks binding [81].

α-synuclein provides not only its own aggregation, but also the aggregation of other amyloidogenic proteins, such as tau and Aβ. This ability to form neurotoxic tau protein oligomers due to synuclein oligomers has been shown in vitro studies [82]. The aggregation of α-synuclein is also induced by tau and Aβ oligomers [83].

For us it is interesting what kind of oligomer structures will be predicted if to use AlphaFold2 in two different regimes [53]. Figure 9 demonstrates α-synuclein predictions using two approaches of the AlphaFold2 prediction program (AlphaFold2 Colab and AlphaFold2 advanced). Interestingly, according to these predictions, the tetramer differs little from the trimer. At the same time, as shown by experimental data and simulation results, it is the tetrameric structure that is the native and widespread structure of α-synuclein, which is in dynamic equilibrium with the monomer [84,85,86,87]. It was also demonstrated that the α-synuclein tetramer has a pronounced α-helical structure, and it is possible that its destruction leads to the pathological aggregation of this protein. When using AlphaFold2 Colab, we obtained the prediction by adding molecules to each other, whereas when using AlphaFold2 advanced, the prediction was made by simple oligomerization. In both cases, when using six molecules of α-synuclein, we see the formation in the form of a “pore”, similarly when using eight molecules. However, when ten or more molecules are predicted, models with no similarity are formed. When using the AlphaFold2 Colab modification, the resulting hexamer structure is more compact, because it contains fewer unstructured regions; in turn, the structure predicted by AlphaFold2 advanced has a greater number of unstructured regions. When predicting the interaction of eight molecules, the structures are almost similar. In the predicted structures of the hexamer and octomer, one can see the resulting “pore” in the form of α-helices. Based on the predicted structures, it can be concluded that both variants of the prediction take place and the possibility of using them to perform cloning of truncated sections can show a more stable structure.

### 3.3. Insulin

The discovery of insulin in 1922 was a transformative event in the development of molecular medicine. Preparations of insulin and its analogues, entering the body, in addition to the direct therapeutic effect in the treatment of diabetes mellitus, can provoke undesirable effects, which are known as insulin amyloidosis and toxic effects on cells at injection sites [88,89]. The formation of insulin amyloid fibrils is a well-studied phenomenon that has biological and pharmaceutical implications. It is known that heating of protein solutions is used in model systems in order to induce their fibril formation [90]. At the same time, improper storage and transportation of insulin preparations (at room temperature and above) also provokes insulin fibril formation [91]. The aggregation of insulin prevents rapid uptake of insulin by cells, which in the treatment of diabetes can lead to localized injectable amyloidosis, which provokes an undesirable immune response and tissue necrosis. Insulin amyloidosis reduces the stability and activity of pharmaceutical compositions of insulin preparations and its analogues, aggravates the course of diabetes mellitus, increases the cost of the treatment procedure and makes it less accessible [92].

The facts of injectable amyloidosis, which tend to increase in recent years, accompanying the treatment of patients with diabetes mellitus with insulin and its analogues, have been established [88]. Modern insulin preparations usually contain phenol and its derivative, metacresol (hereinafter referred to as “phenols”), which are used as inhibitors of the formation of amyloid fibrils and antimicrobial agents. At the same time, the toxic effect that phenols have on human connective tissue cells (fibroblasts) at injection sites of insulin or its analogue is known [89]. Apparently, phenols are ineffective as inhibitors of the amyloid formation of insulin and its analogues, which is confirmed by the increase in clinical cases of injectable insulin amyloidosis [93]. Thus, an urgent scientific task is to find the most effective and least toxic substitutes for phenols in insulin preparations and its analogues.

A fairly large amount of experimental material on potential inhibitors of insulin amyloid formation has been published in the scientific literature [94,95,96]. However, such studies do not consider specific substitutes for phenolic compounds in insulin preparations; therefore, the testing of new low-toxic molecules that can inhibit the formation of amyloids and have an antimicrobial effect is of great scientific importance for the development of new insulin preparations and its analogues that do not contain toxic phenols.

In the course of implementing one or another strategy for the development of low-toxic inhibitors of amyloid formation, it is necessary to solve the following tasks. Using bioinformatics tools and docking programs it is possible to search for promising inhibitors of insulin amyloid formation and its analogues. Some researchers pay special attention to computational methods, in particular, molecular dynamics, noting their importance for elucidating the mechanism of amyloid formation, as well as creating new insulin analogues [97]. Based on a theoretical analysis, it is necessary to select and synthesize peptide inhibitors of the amyloid formation of insulin and its analogues. Then, it is required to analyze the kinetic curves of changes in the fluorescence intensity of an amyloid-specific dye (for example, thioflavin T) in pairs of “peptide inhibitor of amyloid formation—insulin (or insulin analogue)”. By analyzing the metabolic activity and viability of human fibroblasts, it is possible to study the cytotoxicity of promising inhibitors of insulin amyloid formation and its analogues. In the future, it is possible to identify under in vitro conditions a peptide non-toxic preservative and an inhibitor of fibril formation in insulin preparations, functionally effective for replacing toxic phenols in preparations of insulin.

Thus, the successful solution of the objectives of this strategy will allow us to study and propose effective non-toxic inhibitors of amyloid formation in new insulin preparations. In turn, insulin preparations with improved properties, in particular, more stable during long-term storage and with fewer side effects, will be used in the treatment of insulin-dependent diabetes mellitus.

Insulin is a good model for studying the process of amyloid formation, and this process has been studied since 1947. It has been shown that after fibril breakdown, insulin remains functional, which means that there are no major structural rearrangements during fibril formation. Our data also showed that the building block for building fibrils is the oligomer, and the stacking of these oligomers is clearly visible at the initial stages of kinetics on electron microscopy images (Figure 5). Studying the amyloid formation of insulin and its rapid acting analogue lispro and the long-acting analogue glargine, we showed that all preparations form fibrils according to the exponential growth mechanism (branching), with the size of the primary nuclei being one monomer and the size of the secondary nuclei being 0 [98]. In contrast to the Aβ fibrils, insulin fibrils do not have as much polymorphism. However, they tend to form large structural fibril clusters [31]. Figure 10 demonstrates the spine of insulin fibrils obtained in the experiments with limited proteolysis and mass spectrometry analysis of protein fragments produced by limited proteolysis [99,100].

## 4. Influence of Various Molecules on the Aggregation Process

### 4.1. Inhibitors for Aβ Fibril Formation

Several approaches have been proposed to create effective inhibitors of β-secretase activity in humans, mainly using peptidomimetics, which are non-convertible analogues of BACE1 substrates. There are many studies on β-secretase inhibition, including the docking of a number of flavonoids [101], as well as a number of virtual screening studies [102,103] that revealed the causes of inhibition of this enzyme [104]. Studies of BACE1 mutants in mice indicate that there may be serious side effects when this particular enzyme is inhibited. In particular, one of these effects may be neurodegeneration, which is a serious problem [105]. As for γ-secretase, this intramembrane protein is involved not only in APP proteolysis, but also in several other processes [106]. These data make us treat the creation of inhibitors of these enzymes with some caution. Taking into account the involvement of APP in synaptogenesis and the functioning of the nervous system, this approach to the treatment of AD seems unpromising, since a decrease in the expression of APP and related proteins can lead to undesirable side effects.

Something that was quite unexpected, but predictable from a biochemical point of view, was the discovery that the activity of β-secretase is regulated by constitutive cellular proteins PrPc prions, modified forms of which (PrPsc) are a pathogenic factor in the development of Creutzfeldt–Jakob disease and mad cow disease [107]. This indicates a link between Aβ and PrPc metabolism. In the normal brain, the latter regulates the activity of β-secretase, preventing the accumulation of Aβ. Violation of the synthesis of PrPc or its conformation in the case of Creutzfeldt–Jakob disease or mad cow disease leads to the accumulation of amyloid peptide, which is also a characteristic feature of these diseases [108].

It has been shown that a number of low molecular weight natural products, such as polyphenols (resveratrol, quercetin) inhibited Aβ amyloid aggregation [109,110]. Cuccioloni et al. demonstrated that organoruthenium derivative of curcumin (RuCurcumin) inhibited the formation of Aβ aggregates and also protected nerve cells from the toxic effects of amyloids [111]. Fu et al. demonstrated that curcumin and resveratrol block the oligomerization process by interacting with the N-terminus (residues 5–20) of Aβ(1–42) oligomers along with disruption of the preformed Aβ aggregates [109]. Additionally, curcumin and doxycycline can inhibit the aggregation and hinder fibril elongation of Aβ(1–42), according to data of molecular dynamics (MD) simulations [112]. Another group showed that epigallocatechin-3-gallate (EGCG) can directly bind aggregates and remodels mature Aβ fibrils into less toxic assemblies [113] (Table 1). It has also been shown by molecular dynamics, best inhibitory potential of EGCG and curcumin on unfolded amyloid Aβ(1–40) fibrils [114]. Gallic acid has been shown to reduce Aβ(1−42) aggregation, and also to directly prevent the formation of toxic oligomers and fibrils [115]. However, the hydrophobic property of polyphenols (curcumin, quercetin and other) and possible biomodifications limits their applications [116,117].

Interestingly, multidisciplinary approaches that combine computational and high-throughput experimental methods are expected to become increasingly useful for development non-toxic inhibitors Aβ aggregation [118]. Peptide-based inhibitors derived from original amyloid sequences may advance develop an effective treatment strategy against AD [119]. In fact, the peptide-based amyloid inhibitors have special properties, including low toxicity and side-effects, high selectivity, low accumulation in tissues, which can further be rationally designed and synthesized [120]. Previously, peptide inhibitor KLVFF segment of Aβ have been identified to bind to a hydrophobic Aβ sequence and modulate toxic protein aggregation [121]. In addition, conjugation or adsorption technique can be applied on the simple surface interface interactions between Aβ and AuNPs. Among the nanoparticles, gold nanoparticles (AuNPs) have reported low cytotoxicity. Therefore, Xiong et al. developed a kind of dual peptide (VVIA and LPFFD) coupled AuNPs for inhibiting Aβ aggregation and amyloid cytotoxicity [122]. Recently, Stark et al. identified a peptidomimetic (compound C1) starting from the retro-inverso analogue of KLVFF as an inhibitor of Aβ(1–42) aggregation [123]. The novel Aβ aggregation modulator GAL-201 exhibits high binding affinity to Aβ(1–42) monomers (KD = 2.5 ± 0.6 nM) [124]. Robinson et al. studied the mechanisms of pseudopeptides inhibitors of Aβ(1–42) aggregation, and some advantages compared with monoclonal antibodies have been demonstrated [125]. Aβ/IAPP (islet amyloid polypeptide) cross-interactions have exploited to design Aβ amyloid core mimics (ACMs) as peptide inhibitors of amyloid self-assembly Aβ(1–42) [126]. The finding of novel anti-amyloidogenic synthetic polymers will offer a ground-breaking push for developing anti-amyloid drugs. As demonstrated, anti-amyloidogenic polymeric materials bind to the natively unfolded amyloidogenic proteins and peptide and redirects the aggregation pathway toward off-pathway, non-toxic assemblies, which blocks fibrillization [127]. Previous studies have revealed that incubation of Aβ peptides with polyacrylamide-based glycopolymers containing β-d-glucose pendant groups have an inhibitory effect on Aβ aggregation [128]. In another recent article, Evgrafova et al. have demonstrated thermoresponsive polymers varying in hydrophilicity/hydrophobicity ratio to selectively modulate Aβ aggregation kinetics [129]. Monoclonal antibodies (mAbs) directed against Aβ (bapineuzumab, crenezumab, solanezumab, and aducanumab) caused a lower incidence of AD [130]. However, increasing dosage in the later period of AD might have a negative impact on the trial results; in particular, antibodies may show off-target cross-reactivity [131].

### 4.2. Inhibitors for α-Synuclein Fibril Formation

In the fight against neurodegenerative diseases associated with the deposition of α-synuclein aggregates in the brain, oligomer inhibitors are used as the main toxic forms. The inhibitory effect of various peptides, antibodies, and small molecules on the process of α-synuclein aggregation was shown due to the stabilization of the molecule. We mentioned above that the rate of fibrillation depends on the number of open ends of the molecules to which new monomers or oligomers are added. Therefore, when access to the ends of the molecule is closed, the process of fibrillation is suppressed. For this purpose, α-synuclein dimers can be used as the most efficient method [49].

The first described inhibitors of α-synuclein aggregation were catecholamines: dopamine and L-DOPA. It was shown that under the influence of these compounds, the disassembly of α-synuclein fibrils occurs with a decrease in the size and number of molecules in vitro. Studies on mice have shown similar results. The addition of catecholamines by the Michael reaction, the formation of the Schiff base, leading to the modification of α-synuclein, are considered as the main mechanism of disaggregation. However, oligomers remain that exhibit greater cytotoxicity than mature fibrils [132].

DOPAC (3,4-dihydroxyphenylacetic acid), a normal product of dopamine metabolism, also inhibits fibrillation by stabilizing oligomeric and protofibril forms of α-synuclein. Under physiological conditions, DOPAC is oxidized to DOPAC-quinone, which is directly involved in the reaction. Complete inhibition of fibrillation occurs at a ratio of acid to protein of 1:1 or higher. In this case, aggregation also stops at the oligomer stage [133].

Dopamine analogues, hydroquinone and catechol, stop the aggregation process, but with a greater proportion of dimers and monomers than cytotoxic oligomers, as in the case of dopamine, which is a more beneficial solution [134].

Quinones spontaneously form when polyphenols are oxidized by oxygen dissolved in water, so polyphenols are also used as inhibitors of α-synuclein aggregation. The main mechanisms of inhibition are associated with the covalent interaction of polyphenolic compounds with tyrosine or lysine, or the oxidation of methionine under the influence of compounds. Polyphenols are natural compounds, plant metabolites. They are widely studied compounds for stopping fibrillation in vitro. Their undoubted advantages are their prevalence, availability, non-toxicity. Substances contained in tea, coffee, and spices are being studied.

Epigallocatechin-3-gallate (EGCG) is one of the best-known inhibitors of α-synuclein aggregation, a polyphenol found in green tea. There is a model showing that EGCG binds to α-synuclein at three sites and rearranges stable fibrils into disordered structures by disrupting intrapeptide hydrogen bonds [135]. Mass-spectrometry-controlled H/D exchange has been shown to decrease the toxicity of oligomers when EGCG is added by reducing the binding of cell membranes to the N-terminus of the protein and facilitating membrane permeabilization [136]. At physiological pH, EGCG is oxidized and has a greater inhibitory activity compared with the reduced form, but the inducing ability of primary nucleation is enhanced. When environmental conditions change (solution pH, presence and absence of nuclei, reaction conditions), the effect of epigallocatechin gallate on α-synuclein fibrillation changes to fibrillation acceleration [137].

Another well-known agent used is curcumin (Table 1). Curcumin is a natural polyphenol that has the ability to inhibit the pathological aggregation of α-synuclein and the ability to disaggregate formed fibrils. Several potential binding sites for curcumin are located in the central NAC domain of α-synuclein. The effect of curcumin is based on the disruption of long-range interactions within the chain, which allows it to reconfigure more quickly [138]. During phase separation, curcumin and α-synuclein interact more easily due to hydrophobicity. This interaction with condensates reduces the diffusion of the protein, which ensures the inhibition of the formation of aggregates. Curcumin has also been shown to inhibit not only wild-type (WT) amyloids, but also α-synuclein mutant E46K and H50Q amyloids in condensates, making it a potential treatment for PD [139]. In addition to the above, there is an alternative hypothesis about the mechanism of inhibition of amyloid aggregation, which consists of inhibiting the generation of oxidative stress, increasing reduced glutathione, and preventing a glial-associated inflammatory response [140]. Despite the rather obvious biological effect of curcumin on α-synuclein aggregates, it has a number of disadvantages that prevent its widespread use in practice. These disadvantages are low water solubility, poor bioavailability [141].

The “half” of the curcumin molecule is cinnamon and their derivatives. Due to one pharmacophore group, the compounds also have the potential to prevent fibrillation of α-synuclein. Unlike curcumin, these compounds are water-soluble, allowing for greater bioavailability. Cinnamic acid derivatives found in plant foods have anti-amyloid transformation activity. In particular, the inhibitory effects of 3,4-dimethoxycinnamic acid, 3-methoxy-4-acetamidoxycinnamic acid, and ferrulic acid have been shown in vitro studies [142,143]. Derivatives of hydroxycinnamic acids, unlike curcumin, interact only with structured proteins; therefore, they cannot interact with the monomeric form of α-synuclein due to its disorder. However, they bind to oligomers and fibrils and prevent further aggregation [144]. The exact mechanism of inhibition has not yet been studied; it is assumed that aromatic rings interact with α-synuclein, creating steric hindrances for further fibril growth [145].

Another class of compounds considered as potential inhibitors of the amyloidogenic transformation of α-synuclein are alkaloids. They are secondary metabolites of plants and have properties that help with neurodegenerative diseases: reduce neuroinflammation, enhance neuroprotection [146].

Caffeine is an alkaloid found in many beverages such as tea, coffee, and energy drinks. Studies have shown that caffeine accelerates the aggregation process due to the formation of altered α-synuclein molecules, which, upon aggregation, form amorphous oligomers that are less toxic than fibrillar ones [147]. In a study on the A53T mutant strain, caffeic acid was shown to promote protein degradation by enhancing cell autophagy, but no effect was found for the wild type [148].

In a recent study, a team of authors also tested other plant extracts: synephrine, trigonelline, cytisine, harmine, cumine, gupechenin, and pamisin, for their ability to inhibit α-synuclein fibrillation. These compounds have shown positive results at various molar ratios [149].

The study of the action of small molecules on α-synuclein is also of interest to scientists. The currently studied aromatic compound SynuClean-D, which is 2-hydroxy-5-nitro-6-(3-nitrophenyl)-4-(trifluoromethyl)nicotinonitrile, has been shown to inhibit WT and PD-associated mutants H50Q and A30P, and also destroy newly formed amyloid fibrils [150]. In subsequent work, SynuClean-D showed the ability to reduce the accumulation of α-synuclein phosphorylated aggregates [151].

Another small molecule inhibitor, the isoquinoline derivative Fasudil, approved for clinical use in humans, binds to tyrosines at the C-terminal region of monomeric α-synuclein (Y133 and Y136), which delays aggregation [152].

The similarity of these molecules that inhibit the fibrillation of α-synuclein is that they have a flat hydrophobic aromatic core, from which polar branches extend, which do not allow the creation of hydrogen bonds between α-synuclein molecules. Recent studies have shown that only one aromatic ring is needed to inhibit aggregation [153].

Aptamers are single-stranded short fragments of nucleic acids that specifically bind to a target. Several studies have shown that the in vitro and in vivo addition of aptamers to the prefibrillar structures of α-synuclein reduces their aggregation, preventing monomers from attaching to the ends. This improves neuronal survival [154]. The attachment of the aptamer probably occurs to oligomers. Thus, oligomers with which the aptamer did not bind follow the usual path of transformation into fibrils. Additionally, the aptamer–α-synuclein complex induces the formation of aggregates through an alternative, yet unknown, pathway [155].

The problem with most fibrillation inhibitors is that these compounds prevent the formation of mature fibrils, but stabilize the oligomeric state of α-synuclein, which has the greatest cytotoxic effect. Therefore, the search for molecules that would inhibit the oligomerization process and also disaggregate the resulting oligomers to a monomer should be continued.

### 4.3. Inhibitors for Insulin Fibril Formation

Elimination of side effects of drugs based on proteins and peptides is a modern area of scientific research in molecular biology and medicine [156,157]. In our opinion, the main directions of research on this problem are related to obtaining experimental data on the efficacy and safety of new candidate molecules. Traditionally, phenols, in addition to the effects of aggregation inhibition, provide sterility and a long shelf life of insulin preparations [158]. At the same time, a significant amount of scientific research is aimed at finding new substitutes for toxic components of medicinal preparations. Despite the fact that phenolic components are commonly used in commercial insulin preparations as fibril inhibitors and antimicrobial agents, only recently have individual research teams noted their negative effects [89,159,160]. In particular, components of insulin preparations such as phenol and metacresol have been shown to induce the expression of inflammatory cytokines [159]. Other researchers have demonstrated that metacresol in the composition of insulin preparations interact with cell membrane lipids and provoke the destruction of the phospholipid bilayer [160]. All medical insulin preparations contain mainly metacresol and phenol, and it is these phenolic compounds that are responsible for the cytotoxicity of insulin preparations [89]. However, other researchers have not proposed a comprehensive solution for an effective and least toxic substitute for phenols in the preparations of insulin and its analogues.

Synthetic and natural small molecules and peptides have been studied to inhibit insulin aggregation [161,162,163,164,165]. Some authors pay special attention to the consideration of the peptidomimetic approach for the modulation of important amyloid proteins (Aβ, α-synuclein, insulin, islet amyloid polypeptide and mutant p53) [166]. Unfortunately, many inhibitors cannot be used in medical practice due to potentially high toxicity to human cells. However, peptide inhibitors attract attention, since they can potentially have a lesser toxic effect compared to non-peptide inhibitors [167,168,169]. Short peptides are effective and simple in this regard, especially those derived from the self-recognition regions of amyloidogenic peptides and proteins. For example, Gibson and Murphy reported that a synthetic peptide RRRRRRLVEALYL (where VEALYL is part of amyloidogenic regions B12-17 of insulin) significantly reduces insulin fibrillation [170].

It should be noted that, using peptide inhibitors as an example, the use of bioinformatic tools is a new and promising direction for the prediction and initial selection of active substances with a high antimicrobial effect and low cytotoxicity. Previously, using the FoldAmyloid [171], PASTA 2.0 [172], Waltz [173], AGGRESCAN [174], and ZipperDB [175] programs, we predicted and then experimentally determined amyloidogenic regions in insulin and its analogues (lispro and glargine) [99,100]. Interestingly, the differences in the amyloidogenic regions of insulin, lispro, and glargine were mainly due to the fact that the amyloidogenic core of glargine did not include the B-chain region FVNQH (B1-B5).

The theoretical determination of fibril-inhibiting peptides of insulin and its analogues can be performed using molecular docking tools, for example, AutodockTools [176,177] and CABS-dock [178] (Figure 11).

Using molecular docking, it is possible to calculate the binding energies and inhibition constants of peptide ligands with respect to the amyloidogenic regions of insulin and its analogues. In vitro activity of potential anti-amyloid and antimicrobial molecules can be assessed, in particular, using fluorescence analysis with amyloid-specific dyes (for example, thioflavin T), atomic force and electron microscopy of the presence/absence of amyloids in insulin preparations with amyloid formation inhibitors, determination of the survival of bacterial cells with phenol substitutes. An analysis of the fluorescence kinetics of thioflavin T allows us, in turn, to evaluate the effects of inhibitors on the lengthening of the lag period of fibril formation, and changes in the relative amount of amyloid fibrils [179,180]. Electron microscopy is a direct method for visualizing the formation/destruction of amyloid fibrils [181]. Based on electron microscopic images, changes in the diameter, length of individual fibrils, number, uniformity/heterogeneity of fibrils caused by inhibitors of insulin fibril formation can be established. In addition, electron microscopy makes it possible to detect and distinguish between amorphous and ordered aggregates. It is known that an increase in the concentration of an inhibitor can enhance the effect of reducing fibril formation; however, it is desirable that the concentrations of active substances do not go beyond certain physiological limits and do not cause a cytotoxic effect in relation to eukaryotic cells [182]. At the same time, the antimicrobial effect of new inhibitors can be determined in relation to model (*Thermus thermophilus*, *Escherichia coli*) and pathogenic (*Pseudomonas aeruginosa*, *Staphylococcus aureus*) microorganisms by calculating and experimentally verifying the minimum inhibitory concentration (MIC) of the substance [183,184]. With the using of resazurin or the MTT test, it is possible to identify the cytotoxic effect of selected substances in relation to human fibroblasts [185]. MTT-test allows to evaluate cell viability and metabolic activity of eukaryotic cells treated with potential inhibitors [186]. Data on the cytotoxicity of potential inhibitors of fibril formation in comparison with phenols can be obtained over the course of experiments to assess the survival of fibroblasts in vitro. The antimicrobial activity of phenol substitutes in insulin formulations can be compared to both conventional antibiotics and new antibiotics such as antimicrobial peptides [187]. Thus, it is possible to design and synthesize new peptide inhibitors of amyloid formation [188] with properties of antimicrobial peptides [185].
ijms-24-03781-t001_Table 1Table 1Common inhibitors for Aβ, α-synuclein, and insulin.Aβα-SynucleinInsulin
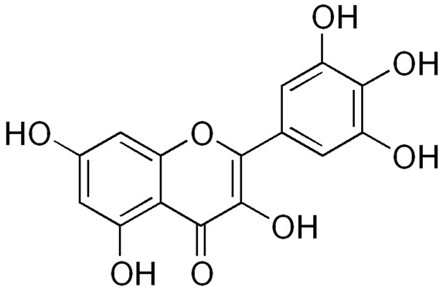
Myricetin10 µM, decreased of Aβ(1–42) fibril formation (10 µM Aβ solution) [189] 200 µM, inhibited of α-synuclein fibril formation (200 µM α-synuclein solution) [190] 400 µM, inhibited of insulin fibril formation (40 µM insulin solution) [191] 
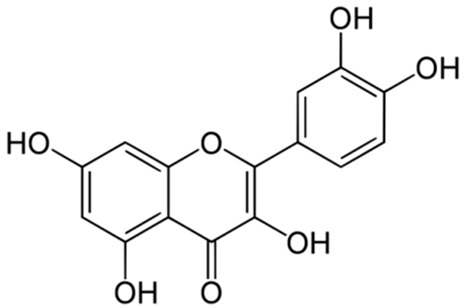
Quercetin50 µM, decreased of Aβ(1–40) fibril formation (50 µM Aβ solution) [110] 20 µM, inhibited of α-synuclein fibril formation (70 µM α-synuclein solution) [192] 85 µM, inhibited of insulin fibril formation (170 µM insulin solution) [193] 
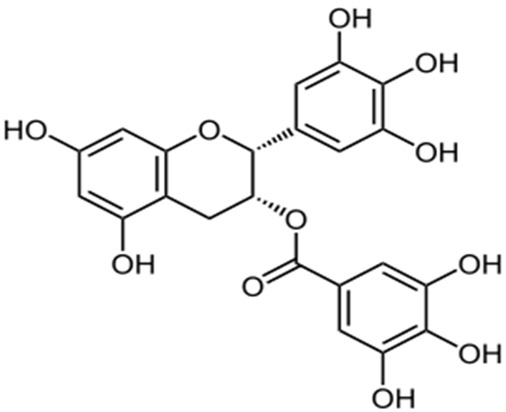
Epigallocatechin-3-gallate25 µM, inhibited of Aβ(1–42) fibril formation (2 µM Aβ solution) [194] 100 µM, inhibited of α-synuclein fibril formation (100 µM α-synuclein solution) [185]25 µM, inhibited of insulin fibril formation (200 µM insulin solution) [185]
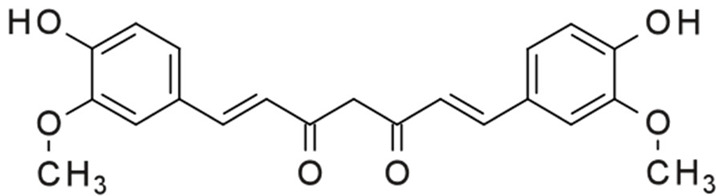
Curcumin1 μM, disaggregated fibrils of Aβ(1–40) (11.6 µM Aβ solution) [195] 72 μM, disaggregated fibrils of α-synuclein (48 µM α-synuclein solution) [138] 4 μM, decreased of insulin fibril formation (3 µM insulin solution) [196] 
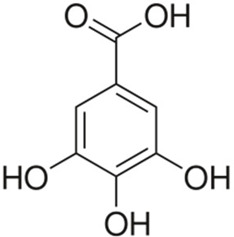
Gallic acid5 μM, reduced the fibril sizes of Aβ(1–42) particles (2.5 µM Aβ) [115] 5 μM, inhibited of α-synuclein fibril formation (20 µM α-synuclein solution) [197] 1700 μM, reduced insulin fibril formation (340 µM insulin solution) [198]


## 5. Conclusions

The study of the mechanisms of pathological protein aggregation, the search for potential agents capable of inhibiting the process of fibrillation and activating the process of their disaggregation, is a hot topic in our time. We hope that modern approaches will make it possible to develop and synthesize effective non-toxic inhibitors of amyloid fibril formation. As a result, peptides or a set of inhibitors will be created that prevents the formation of amyloid fibrils of Aβ, α-synuclein, insulin and its analogues. Thus, a significant step can be taken towards solving the problems associated with the toxic effect on human cells, both of the amyloids themselves and of potential inhibitors that can be used in medical practice. New non-toxic inhibitors of amyloid formation can be used for the future rational development of new drugs that do not contain toxic components.

## Figures and Tables

**Figure 1 ijms-24-03781-f001:**
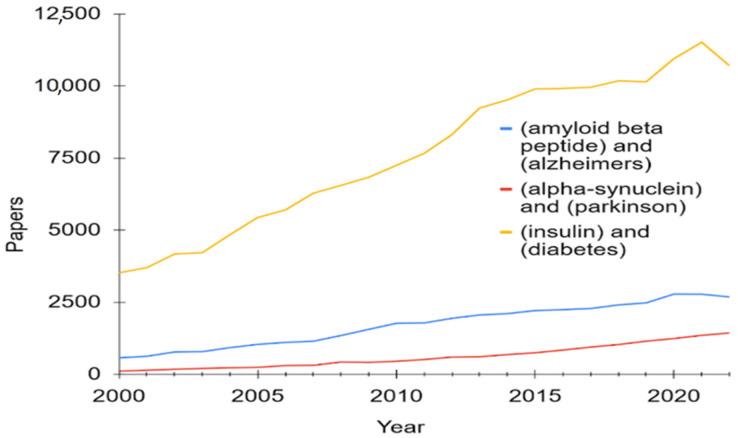
PubMed statistics for Aβ, α-synuclein, and insulin and diseases associated with each of these proteins.

**Figure 2 ijms-24-03781-f002:**
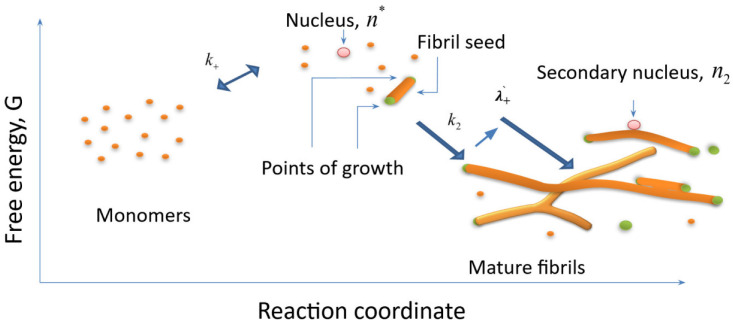
Free energy change during initiation and subsequent exponential fibril growth in the case of branching scenario, *n** is the primary nucleus size, *n*_2_ is the secondary nucleus size.

**Figure 3 ijms-24-03781-f003:**
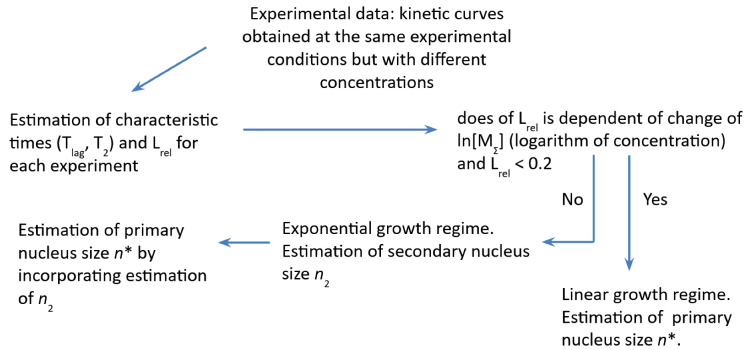
General scheme of experimental data analysis of kinetics of amyloid fibril formation.

**Figure 4 ijms-24-03781-f004:**
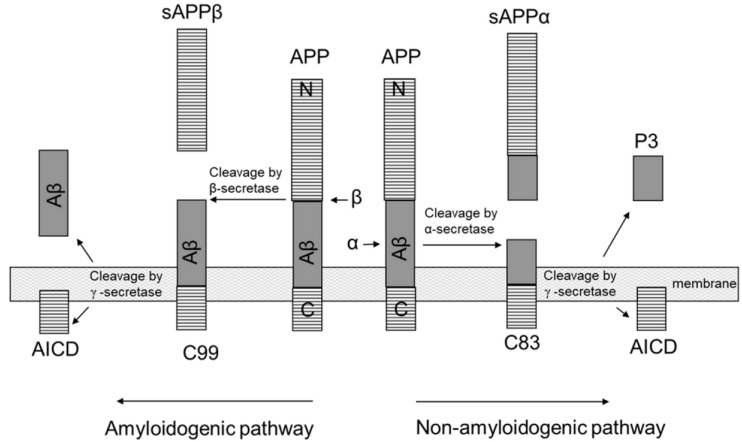
APP cleavage scheme in two ways.

**Figure 5 ijms-24-03781-f005:**
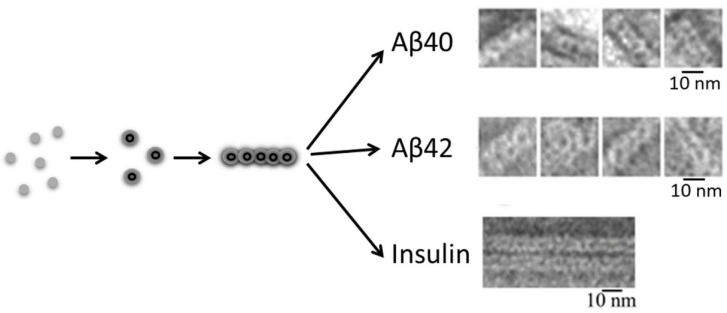
Schematic representation of the process of fibril formation by the recombinant Aβ(1–40), Aβ(1–42) peptides and insulin. Aβ(1–40) and Aβ(1–42) fibrils are formed under conditions of 50 mM Tris-HCl (pH 7.0–7.2) [16] and insulin at 37 °C, 20% acetic acid (pH 2.0), 140 mM NaCl [31]. The main structural element of insulin fibrils is a ring-like oligomer of about 6–7 nm in diameter, about 2–3 nm in height and about 2 nm in diameter of the hole.

**Figure 6 ijms-24-03781-f006:**
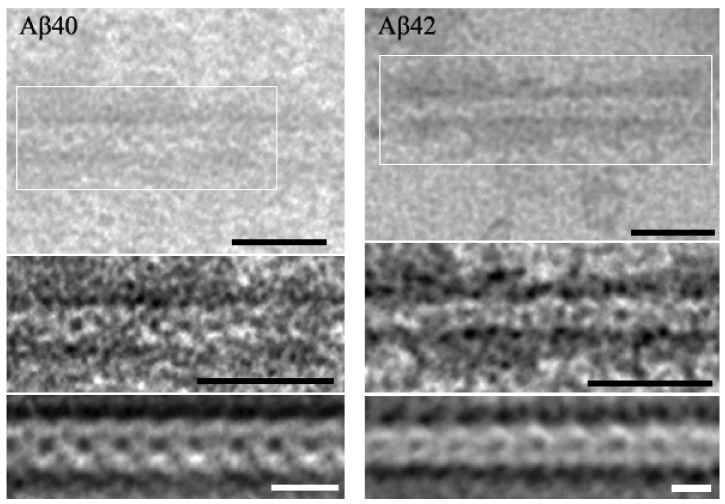
Markham’s method applied to the amyloid fibrils of Aβ(1–40) and Aβ(1–42) formed under conditions of 50 mM Tris-HCl (pH 7.0–7.2). Black labels correspond to 25 nm and white ones to 10 nm.

**Figure 7 ijms-24-03781-f007:**
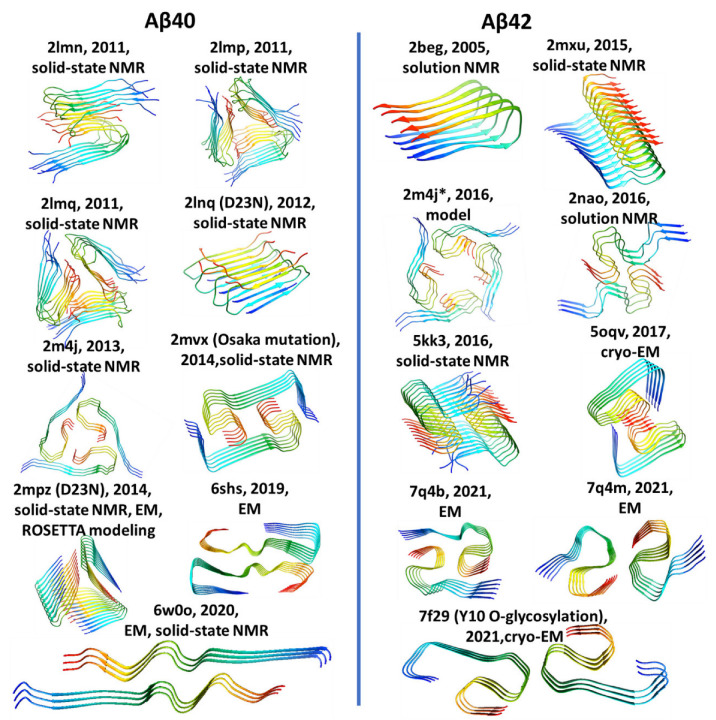
Some structures of Aβ(1–40) and Aβ(1–42) fibrils from the Protein Data Bank. The asterisk denotes the following: a trimer was taken from structure 2m4j and structure 2m4j* was built on its basis. Aβ(1–40) (2lmn—[39], 2lmp and 2lmq—[40], 2lnq—[41], 2m4j—[42], 2mvx—[43], 2mpz—[44], 6shs—[35], 6w0o—[36]). Aβ42 (2beg—[45], 2mxu—[46], 2nao—[47], 5kk3—[48], 5oqv—[32], 7q4b and 7q4m—[37], 7f29—[38].

**Figure 8 ijms-24-03781-f008:**
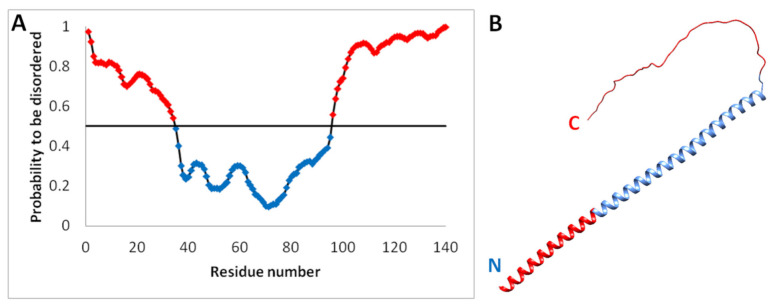
Probability amino acid residues to be disordered for α-synuclein calculated using the IsUnstruct program [52] (**A**). Structure of α-synuclein predicted by the AlphaFold2 program [53] (**B**). Disordered regions are highlighted in red according to the IsUnstruct program.

**Figure 9 ijms-24-03781-f009:**
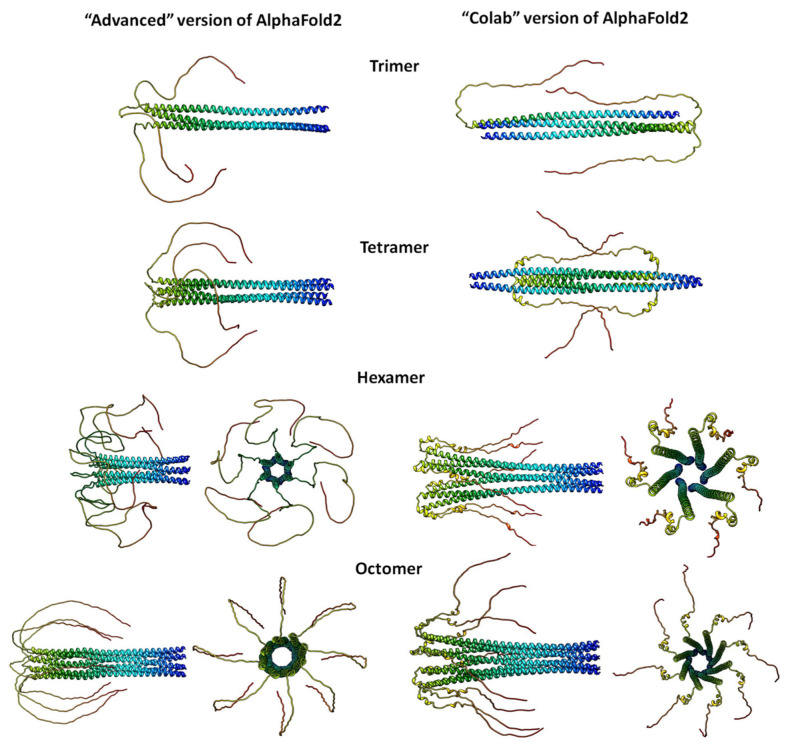
Predicting α-synuclein structures using the AlphaFold2 program. The predictions are made by the “advanced” version of AlphaFold2, which modifies the original Deepmind notebook (both homo- and hetero-oligomers) (left column). The predictions are made by the “Colab” version of AlphaFold2, this notebook does not use templates (homologous structures) but uses a selected part of the Big Fantastic Database (BFD) (right column). While the accuracy is almost identical on many targets, a small part is somewhat different due to the smaller multiple sequence alignments (MSA) and lack of templates. For greater reliability, we recommend using the full open-source version of AlphaFold2 or the AlphaFold2 Protein Structure database.

**Figure 10 ijms-24-03781-f010:**
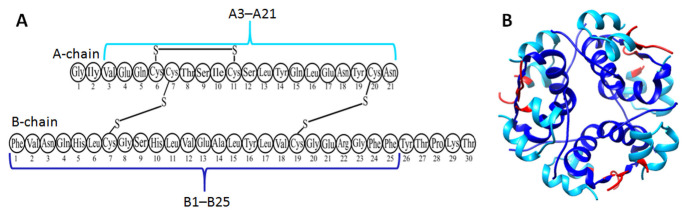
Results of proteolysis of insulin fibrils for amino acid sequences (**A**) and for 3D structure (**B**). PDB file 5e7w was used. Two fragments A3–A21 and B1–B25 are protected (inside the fibrils) from the action of proteases—trypsin, chymotrypsin, and proteinase K. In (**B**), residues accessible to proteases are colored red.

**Figure 11 ijms-24-03781-f011:**
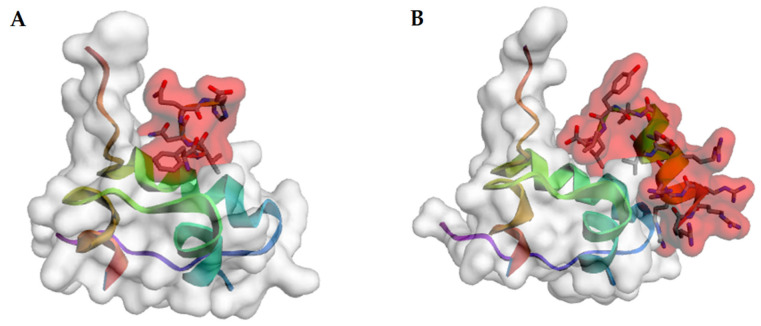
Docking model of insulin monomer (PDB file 1guj) with peptide FVNQH (**A**) and with peptide RRRRRRLVEALYLV (**B**), obtained using the CABS-dock (CABS-dock. Available online: http://biocomp.chem.uw.edu.pl/CABSdock (accessed on 17 December 2022)). The peptide region is shown in red.

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
