# Peer review of "The Strategies of Development of New Non-Toxic Inhibitors of Amyloid Formation"

_ijms, 2023, doi:10.3390/ijms24043781_

Round 1

Reviewer 1 Report

The review by Galzitskaya et al. is a nice and comprehensive account of the state-of-the-art in the field of amyloid research through proteins Aβ, α-synuclein, and insulin with special emphasis on strategies for the development 28 of effective and non-toxic inhibitors of amyloid formation. I recommend publication after the one point has been addressed: When discussing structures of oligomers/multimers of alpha-Synuclein, the authors show AlphaFold predicted helical tetramer structures in Fig. 9 (page 12). In this context, the authors should cite few of the recent experimental and modelling works on alpha-synuclein tetramer and discuss the status of the proposed tetramer as the native state in dynamic equilibrium with monomer:

- Bartels, Tim, Joanna G. Choi, and Dennis J. Selkoe. "α-Synuclein occurs physiologically as a helically folded tetramer that resists aggregation." Nature 477.7362 (2011): 107-110.

- Wang, Wei, et al. "A soluble α-synuclein construct forms a dynamic tetramer." Proceedings of the National Academy of Sciences 108.43 (2011): 17797-17802.

- Xu, Liang, Shayon Bhattacharya, and Damien Thompson. "Re-designing the α-synuclein tetramer." Chemical Communications 54.58 (2018): 8080-8083.

- Nuber, Silke, et al. "Abrogating native α-synuclein tetramers in mice causes a L-DOPA-responsive motor syndrome closely resembling Parkinson’s disease." Neuron 100.1 (2018): 75-90.

Reviewer 2 Report

This is a well-organized concise review on the possible strategies of developing therapeutic molecules against amyloid-related diseases. However, I would like to point out that an emphasis was put on small-molecule-based and antimicrobial peptide-based strategies and few other possible strategies have been overlooked in this review, such as: 1. Nanoparticle-based therapies including peptide-nanomaterial conjugates, 2. Synthetic peptide and polymer-based strategies. Please see this review: ACS Chem. Neurosci. 2021, 12, 1737−1748. Another important recent review was missed: Chem. Rev. 2021, 121, 4, 2545–2647. In my opinion, these newly emerging strategies will play an important role in the near future for developing anti-amyloid drugs and this review will be substantially more comprehensive if the authors include these strategies, however briefly, into their manuscript. Few examples of the polymer-based strategy: Phys. Chem. Chem. Phys. 2019, 21, 20999; Biomacromolecules 2017, 18, 3359−3366.

Reviewer 3 Report

This review does not make a significant contribution to the development of new threapies for neurodegenerative diseases. For example, insulin effects on amyloid formation has been studied since 1947. Do we now expect to develop a new drug using insulin ??? Or other usual substances like resveratrol, curcumin or herbal compunds mentioned in this work??

Reviewer 4 Report

Review of a manuscript “The Strategies of Development of New Non-toxic Inhibitors of Amyloid Formation” by Galzitskaya and coauthors submitted to IJMS.

Accumulating amyloid proteins is a key step leading to the development of many neurodegenerative and other diseases. The study of molecular mechanisms causing these disorders and the search for inhibitors of amyloidogenesis has a paramount significance for contemporary medicine and biomedical area. The progress in these studies will allow us to find new medications to treat these devastating diseases. The manuscript combines the data on three amyloidogenic peptides and proteins: beta-amyloid, α-synuclein, and insulin, and analyzes strategies for the development of efficient inhibitors of amyloid formation. This is a very important biomedical area, and the results and hypothesis of the manuscript will be interesting for the readers of IJMS.

The following corrections and additions should be made.

Abstract

Line 28: “…we will analyze existing and prospective strategies for the development of effective and non-toxic inhibitors of amyloid formation” The main aim of the review described in the abstract is broader than “development of effective and non-toxic inhibitors of amyloid formation. The development of non-toxic inhibitors…” The authors should add and strengthen that it considers amyloid fibril formation mechanisms and fibril formation.

Introduction

Lines 41-43:” Particularly relevant in this regard is the study of protein misfolding, which leads to the formation of pathogenic amyloid fibrils, with the deposition of which in tissues is currently associated with more than 30 human diseases.” After this, the authors should add a reference to :

Uversky VN. The protein disorder cycle. Biophys Rev. 2021 Nov 3;13(6):1155-1162. doi: 10.1007/s12551-021-00853-2.

Line 50. “The second most common is Parkinson's disease.” This should be corrected as “The second most common neurodegenerative disease is Parkinson's disease.”

Line 55:”Epidemiological, clinical and experimental evidence suggests an association between type 2 diabetes mellitus and Alzheimer's disease (AD).”. The authors should add a reference after this sentence: Caveolin: a new link between diabetes and Alzheimer’s disease. Cell Mol   Neurobiology, 2020 Jan 23. doi: 10.1007/s10571-020-00796-4

Lines 66-79 The details about insulin properties and aggregation will be more appropriate in the 3.3 Insulin Section than in the Introduction.

Lines 120-121: Figure 1 demonstrates the PubMed statistics for 3 amyloidogenic proteins considering in this review.” The sentence should be corrected as follows:” Figure 1 demonstrates the PubMed statistics for 3 amyloidogenic proteins considered in this review.” After this sentence, concluding the Introduction the authors should give a short sentence summarizing the aim of this review.

Figure 1. Parts A and B look practically identical. It is not clear what is the rationale for showing two similar parts.  

Lines 189-190: ”The Aβ peptide, which is a polypeptide with a molecular weight of about 4 kDa and consisting of 40-42 amino acid residues, is formed from the Amyloid Precursor Protein (APP)” A reference should be given here.

Line 309: ”Parkinson's disease (PD), dementia with Lewy bodies, Alzheimer's disease (AD)”

Abbreviations for PD and AD are given before, beginning from the Abstract.They should not be repeated after first abbreviation.

 Conclusion

Line 753 -754: “We hope that modern approaches will make it possible to develop and synthesize effective non-toxic peptide inhibitors of amyloid fibril formation.”

It is unclear why the authors mention here only peptide inhibitors, whereas other types of inhibitors have good perspectives.  

Overall, interesting review containing new data and hypothesis.

Round 2

Reviewer 3 Report

Maybe such articles shall be written by drug developer?